# Domains of Vulnerability, Resilience, Health Habits, and Mental and Physical Health for Health Disparities Research

**DOI:** 10.3390/bs12070240

**Published:** 2022-07-18

**Authors:** Rebecca M. Wolfe, Katie Beck-Felts, Brianna Speakar, William D. Spaulding

**Affiliations:** 1Department of Psychology, College of Arts and Sciences, University of Nebraska-Lincoln, Lincoln, NE 68588, USA; bspeakar2@huskers.unl.edu (B.S.); wspaulding1@unl.edu (W.D.S.); 2The Psychology Department, College of Science & Mathematics, Rowan University, Glassboro, NJ 08028, USA; beckfe92@students.rowan.edu

**Keywords:** schizotypy, health disparities, vulnerability models, health behaviors, quality of life

## Abstract

Health disparities associated with severe mental illness (SMI) have become a major public health concern. The disparities are not directly due to the SMI. They involve the same leading causes of premature death as in the general population. The causes of the disparities are therefore suspected to reflect differences in health-related behavior and resilience. As with other problems associated with SMI, studying non-clinical populations at risk for future onset provides important clues about pathways, from vulnerability to unhealthy behavior and compromised resilience, to poor health and reduced quality of life. The purpose of this study was to identify possible pathways in a sample of public university students. Four domains of biosystemic functioning with a priori relevance to SMI-related vulnerability and health disparities were identified. Measures reflecting various well-studied constructs within each domain were factor-analyzed to identify common sources of variance within the domains. Relationships between factors in adjacent domains were identified with linear multiple regression. The results reveal strong relationships between common factors across domains that are consistent with pathways from vulnerability to health disparities, to reduced quality of life. Although the results do not provide dispositive evidence of causal pathways, they serve as a guide for further, larger-scale, longitudinal studies to identify causal processes and the pathways they follow to health consequences.

## 1. Introduction

Health disparities are widespread among people with serious mental illness (SMI), resulting in a life expectancy some 20 years shorter than the general population [1,2,3,4,5,6]. *Serious mental illness* (SMI) is a broad classification of a diagnosable mental, behavioral, or emotional disorders characterized by serious and disabling functional impairment that typically limits one or more of an individual’s life activities [7]. The health disparities in SMI are not attributable to the psychiatric disorder itself, but rather to the major causes of death in the general population, including heart and vascular disease, respiratory disease, cancer, and diabetes [1,5,8,9,10,11,12]. Accordingly, the medical and public health interventions most successful in reducing mortality in the general population also show promise in the SMI population, despite them being inequitably provided to this uniquely vulnerable population [9,13,14,15]. These findings have stimulated interest in the relationships between psychiatric disorders and health-related behaviors (physical activity, sleep, diet, smoking, etc.) that are often the targets of preventive interventions in other populations but are at times overlooked in treatment of SMI, since the primary treatment focus in this population tends to be psychiatric in nature. Additionally, problems more characteristic of SMI but suspected of contributing to health disparities, such as vulnerability to stress, sleep disturbance, and social isolation, are of special interest in designing specialized intervention strategies [16,17,18,19,20,21,22,23,24,25,26,27,28,29,30,31,32].

### Background

*Risk and vulnerability research in schizotypy.* The health disparities issue extends beyond people with psychiatric disorders to people with vulnerabilities that create a risk for the eventual onset of a disorder. Of particular interest with respect to SMI are vulnerabilities of genetic origin, generally understood to be endophenotypes; vulnerabilities with other prenatal origins, e.g., from toxic or infectious factors that are present at birth; and those with origins in childhood adversity, such as abuse and neglect. All three disrupt normal development, resulting in impaired cognitive functioning, self-regulation, and socialization. The construct of *schizotypy* has been a focus of research on such vulnerabilities because it is understood to reflect a state of vulnerability to and risk for the onset of schizophrenia in late adolescence and early adulthood [33,34,35,36,37,38].

*Domains of biosystemic functioning.* Four particular domains of biosystemic functioning relevant to health disparities emerged from the existing background literature and provided the framework for guiding instrument selection and organization of this study’s analyses: (1) vulnerabilities present at birth or acquired in early childhood; (2) skills and abilities associated with resilience to psychological and physical stress, acquired in later childhood, adolescence, and early adulthood; (3) psychological and physical health; and (4) quality of life. The domain of resilience logically divides itself into two subdomains: psychological resilience and healthy behavioral habits and lifestyle. Health logically divides itself into three subdomains: physical health, psychological health, and sleep (which has both physical and psychological components). Quality of life logically divides itself into two subdomains: psychological well-being and social connectedness. Within each domain or subdomain, multiple measurable constructs from previous research were entered into factor analyses to construct variables reflecting key sources of variance for further analysis. Eventually, large-scale longitudinal studies of vulnerable populations will be necessary to fully understand all the mechanisms of the effects of psychiatric disorders on health disparities, and in turn, the effects of health on quality of life. Meanwhile, identification of the underlying dimensions of those mechanisms, as revealed by canonical factors of vulnerability, resilience, health, and quality of life, provide important clues about the pathways of particular vulnerabilities toward the real-world consequences we observe.

*Current research gaps in SMI vulnerability.* Most previous research on vulnerability to SMI has been within specific theoretical contexts: on the construct of schizotypy, the role of cognitive impairments, etc. This approach reveals specific etiological processes that link congenital vulnerabilities and vulnerabilities acquired in early childhood to later development, e.g., in mood/emotion regulation and socialization. However, in the real world, overall vulnerability includes multiple causal, mediating, and moderating constructs. Studies of the roles of specific vulnerability constructs in personal and social functioning and health disparities provide only one piece of the total picture at a time.

This methodological problem is compounded by the inevitability of multiple domains of measurement in health disparities research. Physical health is influenced by multiple factors operating at all levels of biosystemic functioning—physiological, cognitive, behavioral, social, and environmental.

*Purpose of study.* The purpose of this study was to create a set of canonical variables which incorporate diverse constructs that have proven useful in previous vulnerability and health disparities research, suitable for studying processes across multiple domains of biosystemic functioning, and across progressive stages of human development. In contrast to research questions about the contributions of specific constructs to health disparities, such as schizotypy and other developmental vulnerabilities, the question addressed by the canonical approach is, “What are the common elements, across multiple paradigms and constructs, that create pathways, from genetic and acquired vulnerabilities, to the behavioral and lifestyle patterns known to expose or protect people from health risks, to actual psychological and physical health, and to the circumstances and experiences that determine a person’s quality of life?” The answers to this question complement the findings from focused studies of particular constructs, to provide a complete picture of the causes of health disparities and their potential solutions.

*Aims/significance.* Joint consideration of factors within these respective domains and subdomains can create a manageable set of operational measures needed to portray an integrated view of processes that lead from vulnerability to pathology, to health disparities, to compromised quality of life. A comprehensive empirical analysis of all the relationships and pathways involved is methodologically feasible, through the application of advanced multivariate modeling methods. Such methods have demonstrated usefulness in studying vulnerability in the general population (e.g., Stinson et al. [39], this issue), but application to relatively small subpopulations, such as people vulnerable to SMI, is made difficult by the requirement for very large subject samples. Additionally, complete validation of the cause–effect relationships that multivariate modeling can reveal ultimately requires longitudinal research designs. The present study represents the initial stage of a large-scale longitudinal project designed for that purpose. The dataset for this study was the “Time 1” component of longitudinal data collection that will continue for several more months. This cross-sectional dataset is not suitable for complete modeling analyses, but it is suitable for the determination of the underlying dimensions within key domains and subdomains pertinent to psychopathology and health disparities. In addition, simple linear multiple regression analyses can provide an initial indication of possible causal relationships across domains, leading ultimately to health disparities and their consequences on quality of life. These analyses generate key hypotheses to guide subsequent modeling analyses capable of validating causal relationships.

Factor analysis and simple linear regression used in this context do not lend themselves to specific hypotheses, as expected for experimental research. Nevertheless, the expected findings can be expressed in quasi-hypothetical terms. In that sense, the hypothesis of this study is that multiple measurement paradigms based on previously validated constructs within a priori domains of biosystemic functioning and measurement reflect underlying common dimensions, whose relationships with common dimensions in other relevant domains can be quantitatively measured. Support of this hypothesis is a step toward a comprehensive understanding of the role of vulnerability to SMI in health disparities and their consequences on quality of life.

*Study population sample.* The subject population of the present study is students attending colleges and universities in the United States of America. College and university students are frequently used in vulnerability and health research, and not only because they represent a convenient and accessible subject population. College populations are relatively homogenous, compared to the general population, with respect to socioeconomic status, living conditions, and levels of personal and social functioning. This is advantageous for studying vulnerabilities in a context less confounded by those factors. Additionally, college students are generally in the middle of the onset risk window for schizophrenia, allowing for observation of vulnerability factors as they are expressed after a full period of child and adolescent development, but before the onset of frank psychosis. There is a surprising degree of variability in measures of vulnerability in this subpopulation. College and university students are not necessarily representative of vulnerable individuals in different settings or contexts; however, important subpopulations parallel various characteristics, vulnerabilities, and resilience factors found outside of college student populations. It is probably not feasible to assemble a subject sample that matches the college and university population in all pertinent demographic and socioeconomic characteristics except for student status. Pathways from vulnerability to health disparities to quality of life may have unique features in college and university students, but the same must be expected of other subpopulations. Both the common features and the unique features are of scientific interest. The present study articulates features of the college and university subpopulation, and further research will articulate the extent to which the features are unique or common with other vulnerable subpopulations. Understanding both the unique and the common features of any subpopulation will be necessary to fully address the problem of health disparities.

## 2. Methods

### 2.1. Participants and Procedures

Data for this study represent the initial stage of a large-scale longitudinal research project. The entire project is under the oversight of the University of Nebraska—Lincoln’s Institutional Review Board. The full longitudinal project includes a series of seven surveys that span five waves of data collection. Continued data collection for the full longitudinal project is currently underway.

The participants are undergraduate and graduate students attending a large public university in the American Great Plains. Recruitment procedures were designed to (1) take advantage of routine mechanisms for recruiting research participants as part of required coursework, and (2) extend catchment beyond course-related mechanisms to include a broader and more representative sample of university students. Participants were recruited through e-mails sent to students who had previously participated in the university psychology department’s research subject pool and indicated interest in other opportunities to participate in research, a series of recruitment e-mails sent across multiple university listservs reaching various student subpopulations, and posting a recruitment solicitation in the university honors program newsletter. A stratified recruitment strategy was employed, emphasizing listservs and other sources expected to disproportionately access individuals with chronic physical and psychological health conditions and disabilities. Compensation was provided as either course credits or Amazon e-gift cards, or a combination of the two, according to each participant’s choice. In addition, participants who completed all phases of data collection were entered into a lottery to win one of four Amazon Fire Tablets.

Participant inclusion criteria were (1) age between 18 and 35, (2) currently living in the United States, (3) able to speak and read English, and (4) have access to the online questionnaires via computer. Participants confirmed their consent online by responding to the IRB-approved consent form, presented at the beginning of the first set of questionnaires. A total of 213 individuals completed the study protocol, with missing data being randomly distributed. Listwise deletion for missing data was used for all factor and regression analyses, resulting in Ns for specific analyses ranging from 213 to 190.

### 2.2. Measures

The measures selected for this study include both reflective and formative measures [40]. Reflective psychometric measures reflect an underlying psychological construct, thereby requiring confirmation of interitem reliability in the form of Cronbach’s alpha. Formative measures are not psychometric, do not necessarily reflect an underlying construct, and are simple cumulative tallies of items not necessarily expected to be intercorrelated—reliability lies in the accuracy of reporting, in this case self-reporting. Interitem reliability is irrelevant.

All reflective measures with 10 or more items used in this study demonstrated “acceptable” to “excellent” reliability in the present study’s dataset, as measured by Cronbach’s alpha of 0.7 or above. For reflective measures comprised of less than 10 items that did not demonstrate acceptable reliability via a Cronbach’s alpha of 0.7 or above, a combination approach of examining Cronbach’s alpha and interitem correlations was utilized, since interitem correlations are a better method of evaluating internal consistency than alpha for scales comprised of few items with correlations exceeding r = 0.15 being deemed acceptable [41]. Inclusion criteria for those measures in this study were a Cronbach’s alpha of at least 0.5, in addition to an interitem correlation of between r = 0.15 and r = 0.50.

The integrity of all the data was protected by items in the data collection protocol that detected inattention or random responding, e.g., items such as “Please choose ‘3’ (Applied to me very much or most of the time) so we know you are still following along.” Participants’ data were excluded from the dataset if during the third baseline survey they (a) missed more than one of the attention-check validity items indicating inattentive or random responding or (b) they missed one of the attention-check validity items indicating inattentive or random responding and their survey duration was under 70% of the average survey completion time.

*Demographics*. A basic demographics survey collected information about participants’ race, ethnicity, sex at birth, age, marital status, socioeconomic background, childhood environmental urbanicity, medical history of diagnosed chronic health condition(s), individual/family psychiatry history, and disability.

*Developmental vulnerabilities domain.* The construct of *schizotypy* consists of a set of traits that are qualitatively similar to features of schizophrenia spectrum disorders and fall along a continuum of severity in normative populations [42]. Schizotypy is considered a vulnerability for schizophrenia spectrum disorders [37]. Like schizophrenia, schizotypy includes three categories of symptoms and related features, positive, negative, and disorganized [38]. Each category can be measured on a continuous dimension of severity, and schizotypy can be understood as a profile of the three dimensions. In addition to genetic and anatomical links, clinical and subclinical expressions of psychosis are associated with psychosocial factors, including urbanicity, poverty, discrimination associated with minority status, immigration, poor parental communication, poor parental care, and various types of childhood adversity, such as abuse, neglect, and bullying [34,43]. Schizotypy is associated with impairment in social competence, rapport with family and friends, interpersonal engagement, social and recreational activity, and occupational and academic functioning [33,36]. Many of the social cognition and social functioning deficits observed in nonclinical groups are thought to predate and predict the onset of the frank disorder [35,44].

A second construct implicated in developmental vulnerability is *emotion regulation*, which denotes abilities that develop in early- to mid-childhood that serve to regulate and control processes in the central and autonomic nervous system associated with emotional reactivity and behavioral activation. Emotion regulation involves both neurophysiological and psychological components, the control of neurophysiological and primitive psychological processes by higher cognitive processes, and is generally understood to lie in the psychophysiological domain of biosystemic functioning. Distress tolerance and alexithymia are operationally measurable constructs implicated in impairment or disruption of the normal development of emotion regulation, and have been extensively studied in the context of psychopathology and vulnerability.

Distress tolerance is a type of emotional regulation: the ability to maintain functioning despite elevated personal distress. Deficits in distress tolerance and related processes are observed in schizophrenia spectrum disorders [45,46,47]. Alexithymia is a decreased or absent ability to label and describe one’s own emotional state [48], which is of interest in clinical research due to its significant relationship with psychopathology and association with social dysfunction [49,50,51]. Elevated levels of alexithymia have been observed in schizophrenia populations [52,53] and individuals high in schizotypal traits [48,50,54].

A third construct that has received much recent attention in vulnerability research is *childhood adversity*. An extraordinary proportion of people with SMI have histories of child maltreatment [55]. People who have experienced maltreatment as children are at elevated risk for psychiatric disorders, including suicidality [56,57,58]. Among people diagnosed with schizophrenia, child maltreatment history is associated with greater disability [59]. This is consistent with links between trauma and psychosis in adults [60] and high-risk children [61]. More broadly, poverty brings a panoply of childhood adversities, and those adversities are found disproportionately in the histories of people diagnosed with schizophrenia [62].

*Developmental domain: Schizotypy*. The 77-item self-reported Multidimensional Schizotypy Scale (MSS [37]) was used to assess schizotypy. The MSS consists of positive, negative, and disorganized subscales that allow it to measure multidimensional schizotypy. The MSS is a newer measure of schizotypy using updated question wording to reliably measure positive, negative, and disorganized features of schizotypy that better capture the heterogeneity of schizotypal presentations. Higher scores across the three domains indicate greater endorsement of the corresponding schizotypal features. The ranges are 0–26 for positive, 0–26 for negative, and 0–25 for disorganized schizotypy. The MSS-77 has demonstrated excellent internal consistency reliability with both Cronbach’s alpha and binary alpha methods, and good construct validity [37].

*Developmental domain: Alexithymia*. The 20-item self-reported Toronto Alexithymia Scale (TAS-20 [63]) was used to assess alexithymia. It includes three subscales, measuring difficulty identifying one’s own feelings, difficulty describing one’s own feelings, and externally-oriented thinking [63]. Participants rate their agreement with items on a 5-point Likert-type scale. Possible scores range from 20 to 100. Higher scores reflect greater degrees of alexithymia. The TAS-20 has demonstrated good internal consistency and test–retest reliability [63]).

*Developmental domain: Distress tolerance*. The 15-item self-reported Distress Tolerance Scale (DTS [64]) was used to assess an individual’s ability to tolerate feelings of emotional distress; it has subscales that capture several dimensions of distress tolerance, including tolerance, absorption, appraisal, and regulation [64]. Participants are asked to think about times they feel upset or distressed and are then asked to rate how strongly they agree or disagree with a given statement on a scale of “1” (strongly agree) to “5” (strongly disagree). Possible scores range from 15 to 75. Higher scores reflect greater tolerance of distress. The DTS has demonstrated good test–retest reliability, internal consistency, and construct validity [64].

*Developmental domain: Childhood adversity.* The 28-item self-reported Childhood Trauma Questionnaire (CTQ [65]) was used to assess lifetime history of childhood physical, emotional, and sexual abuse. Participants are asked to rate how often each statement applies to them on a scale of “1” (never true) to “5” (very often true). Possible scores range from 28 to 140. Higher scores reflect higher experiences of trauma during childhood. The CTQ has demonstrated good test–retest reliability and convergence and discrimination [66].

*Psychological resilience subdomain*. Beliefs, attitudes, and related traits have long been studied as moderators of health in general, and of psychiatric disorders in particular. Loci of control paradigms applied to health, such as beliefs about the importance of health relative to other personal issues and the degree of personal control one can have over one’s own health, provide quantitative measures of those beliefs. People with SMI express beliefs in this domain consistent with poorer health [67] and predictive of a more disabling course of the disorder [68]. Other familiar psychological constructs understood to provide resilience and counteract vulnerability include self-efficacy, self-esteem, self-concept, and a sense of purpose in life. More recently, new constructs, such as mindfulness [69], grit (persistence and personal initiative [70]), and metacognitive awareness [71], have become candidates for providers of resilience in the face of vulnerability and illness.

*Psychological resilience subdomain: Sense of purpose.* The 14-item self-reported Sense of Purpose Scale (SOPS-2 [72]) was used to assess the extent to which individuals believe their lives have purpose. Subscales include awareness of purpose, awakening to purpose, and altruistic purpose. Participants were asked to rate the degree to which they agree or disagree with each statement on a scale of “1” (strongly disagree) to “7” (strongly agree). Scores range from 14–98. Higher scores reflect a stronger sense of purpose. The SPS has shown construct reliability and criterion validity across all three subscales [73].

*Psychological resilience subdomain: Self-esteem.* The 10-item self-reported Rosenberg Self-Esteem Scale (RSE; [74]) was used to assess self-esteem. Participants were asked to rate how much each statement regarding their self-esteem applies to them on a scale of “1” (strongly agree) to “4” (strongly disagree). Total scores range from 10 to 40; higher scores represent lower self-esteem. The RSE has shown excellent internal consistency and test–retest reliability and concurrent, predictive, and construct validity [74].

*Psychological resilience subdomain: Grit.* The 12-item self-reported GRIT Scale (GRIT-12; [70]) was used to assess passion and perseverance for long-term goals. Participants are asked to rate how much each statement applies to them from “1” (very much like me) to “5” (not like me at all). Scores range from 12–60. A higher total score reflects a greater degree of grit. The GRIT-12 has shown predictive and convergent validity and internal consistency [70].

*Psychological resilience subdomain: Self-efficacy.* The 10-item self-reported General Self-Efficacy Scale was used to assess an individual’s perception of self-efficacy. Participants were asked to rate how true each statement is on a scale of “1” (not at all true) to “4” (exactly true), with total scores ranging from 10–40. Higher scores reflect greater sense of self-efficacy. The GSE has demonstrated good internal reliability and criterion-related validity [75].

*Psychological resilience subdomain: Self-Concept clarity.* The 12-item self-reported Self-Concept Clarity Scale (SCCS [76]) was used to assess the extent to which an individual’s self-beliefs are clearly and confidently defined, internally consistent, and temporally stable. Participants were asked to rate the extent to which they agree with a series of statements ranging from “1” (strongly disagree) to “5” (strongly agree). Total scores range from 12–60. Higher scores reflect greater self-concept clarity. The SCCS has demonstrated good interrater reliability and criterion validity [76].

*Psychological resilience subdomain: Mindfulness.* The 39-item self-reported Five Facet Mindfulness Questionnaire (FFMQ [69]) was used to assess trait dispositional mindfulness across five dimensions, including observing, describing, acting with awareness, non-judging of inner experience, and non-reactivity to inner experience. Participants were asked to rate the extent to which 39 statements were generally true to them from “1” (never or very rarely true) to “5” (very often or always true). Higher scores reflect greater mindfulness. The FFMQ has demonstrated good test–retest reliability and internal consistency and a clear factor structure [69].

*Psychological resilience subdomain: Metacognition.* The 30-item version of the self-reporting Metacognitions Questionnaire (MCQ-30 [71]) was used to assess beliefs about worry and people’s awareness of their own thought processes, capturing the individual differences across five important factors in the metacognitive model of psychological disorders, including cognitive confidence, positive beliefs about worry, cognitive self-consciousness, negative beliefs about the controllability of thoughts and danger, and beliefs about the need to control thoughts. Participants were asked to respond to statements from “1” (do not agree) to “4” (agree very much). Higher scores reflect greater metacognition. The MCQ-30 has demonstrated acceptable-to-good test–retest reliability and good internal consistency and convergent validity [71].

*Psychological resilience subdomain: Locus of control.* The 24-item self-reported Internality, Powerful Others, and Chance Scale (IPOCS-24 [77]) was used to assess a person’s perception of internal control, chance, and powerful others. Participants were asked to respond to statements from “1” (strongly disagree) to “6” (strongly agree). Subscale scores range from 0 to 48. Higher scores reflect greater standing in internal control, chance, and powerful others dimensions. The three-factor approach used in this study has demonstrated the best fit for Levenson’s locus of control model [78]. However, the “Internality” subscale failed to meet the reliability criteria for inclusion in this study and was excluded from further analyses.

*Behavioral resilience subdomain*. The domain of “healthy habits” has long been associated with resilience to physical illness. Meanwhile, unhealthy habits, such as smoking, immoderate use of alcohol, poor diet and nutrition, and lack of exercise, are notoriously over-represented in SMI populations, an association which has recently been empirically confirmed [79,80,81,82]. Unhealthy habits and lifestyles are largely unexplored in vulnerable populations, but some findings suggest they are elevated in at least some subpopulations [83].

*Behavioral resilience subdomain: Health attitudes and behaviors.* The 42-item self-reported Lifestyle and Habits Questionnaire-Brief (LHQ-B [84]) was used to assess individuals’ attitudes and health behaviors. The full measure consists of several subscales assessing an individual’s attitudes and behaviors in regard to different areas of life (e.g., physical health, psychological health, social life, and nutrition). For the purpose of this study, three subscales were used to assess attitudes and behaviors across domains: physical health, nutrition, and exercise. Participants were asked to rate how strongly they agree or disagree with a given statement on a scale of “1” (strongly disagree) to “5” (strongly agree). Subscale scores range from 4 to 30. Higher scores reflect greater self-reported health and nutrition. The LHQ-B has demonstrated good test–retest reliability and construct validity [84].

*Behavioral resilience subdomain: Diet.* The 19-item self-reported Dietary Screener Questionnaire (DSQ [85]) was used to capture persons’ dietary habits by asking participants about the different types of food and drinks they ingested over the past thirty days. For each type of food or drink, respondents were asked how often they ate or drank it in the past 30 days. Some questions also asked respondents to specify the type of food or drink they ingested. For the purpose of this study, formative variables were constructed corresponding to the different components of the food pyramid to get a better understanding of participants’ general eating habits across primary domains, including vegetables, fruit, leafy greens, grains, sugary food, meat, and dairy.

*Behavioral resilience subdomain: Smoking and drinking.* The Alcohol and Substance Use Frequency measure (ASUF) was created as a modification of the 7-item self-reported Drug Use Frequency measure (DUF [86]) to include language updates and additional items to better capture potential prescription and recreational substances for the present day. Updates in the ASUF include the addition of several items that assess the frequencies of substances that were not originally included in the DUF (e.g., Kratom, Kava Kava, Delta 8 THC) and the separation of items into sub-substance use categories (e.g., heroin separated from other opioids). Subscale dimensions were also added to assess the biopsychosocial nature of substance use, including the surrounding environment in which the substance was used (e.g., home, party, and concert), and with whom the substance was used (e.g., alone, significant other, friends, family, or strangers). Alcohol consumption items from the Alcohol Use Frequency Measure (AUF) were combined with items from the DUF in the creation of the Alcohol and Substance Use Frequency measure (ASUF). For the purpose of this study, only three items representing formative constructs were included to assess participants’ frequencies of alcohol consumption, nicotine use, and cannabis use. Participants were asked to report their frequencies of use of a series of substances on a scale of “0” (never) to “5” (daily). All nicotine use, including smoking tobacco, smokeless tobacco (including electronic nicotine delivery systems), and other nicotine products (e.g., chewing tobacco and dip) were combined into a single variable assessing the frequency of nicotine use.

*Physical health subdomain*. Assessment of physical health in the general population also requires a broader scope than in clinical populations, to include more than formally diagnosed medical disorders. This involves inventories of the presence, frequency, and severity of physical symptoms, plus measures of the distress associated with the symptoms.

*Physical health subdomain: Physical symptoms distress.* The 33-item self-reported Cohen- Hoberman Inventory of Physical Symptoms (CHIPS [87]) was used to assess the occurrence and disturbance related to physical symptoms. Participants are asked to choose from a scale of “0” to “4” how much each symptom bothered or distressed them in the last two weeks. The total score ranges from 0 to 132. Higher scores indicate greater distress associated with physical health symptoms. The CHIPS has demonstrated reliability and moderate validity [87,88].

*Physical health subdomain: Physical symptoms report.* A 24-item self-reported Physical Symptoms Report was used to assess an individual’s experience of physical health symptoms (e.g., poor appetite or hands trembling). Participants were asked to select the frequency with which they experience 24 physical health symptoms on a scale of “0” (not at all) to “3” (daily). The total scores range from 0 to 72. Higher scores reflect higher experience of physical health symptoms. This construct was created by modifying a health-symptom frequency item from the Alcohol Use Frequency and Drug Use Frequency Measure’s health subsection. The Physical Symptoms Report’s Cronbach’s alpha was 0.848, demonstrating good scale reliability.

*Physical health subdomain: Sickness behaviors.* The 10-item self-reported Sickness Questionnaire (SQ-10 [89]) was used to assess sickness behaviors, or symptoms that typically arise after inflammatory activation. Participants are asked to choose an answer to a series of ten statements assessing common behaviors associated with inflammatory activation on a scale of “0” (disagree) to “3” (agree). The total scores range from 0 to 30. Higher scores reflect greater perceived sickness behavior. Example items include, “I want to keep still,” “I wish to be alone,” “I don’t wish to do anything at all,” and “My body feels sore.” The SQ-10 has demonstrated adequate reliability, validity, and sensitivity to change [89].

*Sleep subdomain*. Sleep problems are of increasing interest with respect to a broad range of psychological ill-health in clinical, non-clinical, and high-risk populations [25,90,91], and especially in adolescents at high risk for schizophrenia. Sleep problems take many forms, including sleep onset difficulty associated with anxiety, stress or rumination, early awakening, nightmares, and chronic insomnia. Assessment requires identification of the quality of the sleep, which includes both the severity of sleep problems and their impacts on daytime functioning.

*Sleep subdomain: Insomnia-specific rumination.* The 20-item self-reported Daytime Insomnia Symptom Response Scale (DISRS [92]) was used to assess insomnia-specific rumination. Participants were asked to indicate their frequencies of engagement in a series of behaviors when they are feeling tired on a scale of “1” (Almost Never) to “4” (Almost Always), with the total scale score ranging from 20 to 80. Higher scores indicate higher levels of insomnia-specific rumination. Example items include, “Think about how hard it is to concentrate”; “Think, I cannot be around people when I’m feeling this way”; and “Think about how tired you feel.” The DISRS has demonstrated excellent reliability and good internal consistency [92].

*Sleep subdomain: Sleep disturbance.* The 7-item self-reported Insomnia Severity Index (ISI-7 [93]) was used to assess sleep disturbance. Participants were asked to rate on a 5-point scale from “0” to “4” their severity of insomnia problem(s), satisfaction with sleep patterns, interference with daytime functioning, noticeability of impairment on quality of life, and worry/distress related to current sleep problems. Scores across ISI items were summed to obtain a total score indicating the level of sleep disturbance experienced over the past two weeks, with higher scores reflecting greater self-reported sleep problems. The ISI has demonstrated good internal consistency, test–retest reliability, and concurrent validity [93,94].

*Sleep subdomain: Sleep-reactivity.* The 9-item self-reported Ford Insomnia Response to Stress Test (FIRST [95]) was used to assess sleep-reactivity, which is the likelihood that an individual will experience sleep disturbance following various stress events/situations. Participants were asked to rate how likely it is following a series of situations for them to experience difficulty sleeping on a scale from “1” (not likely) to “4” (very likely). The total scores range from 9 to 36. Higher scores reflect greater “stress-related” vulnerability to sleep disturbance. The FIRST has demonstrated high reliability and good construct validity [95].

*Psychological health subdomain*. Assessment of overall psychological health in a non-clinical population arguably requires a broader scope than in clinical populations, because of the importance of less acute or more subtle distress that is often eclipsed by diagnosable disorders. Accordingly, an assessment protocol must cover clinical and subclinical levels of mood and emotion dysregulation, anxiety, stress, and cognitive difficulty, including problems with memory and behavioral organization. Of recent particular interest are *attenuated psychotic symptoms*, understood as milder versions of symptoms that would fulfill diagnostic criteria for a psychotic disorder at more severe levels. Attenuated psychotic symptoms may be present in the general population at surprisingly high levels [96], but their presence in vulnerable populations is linked to the onset of psychotic disorders and symptom expression after onset [97].

*Psychological health subdomain: Mood symptoms.* The 21-item self-reported Depression Anxiety and Stress Scales (DASS-21 [98]) was used to assess depression and anxiety. The complete measure consists of three scales looking at depression (e.g., “I felt that life was meaningless” and “I felt that I had nothing to look forward to”), anxiety (e.g., “I experienced breathing difficulty” and “I felt I was close to panic”), and tension/stress (e.g., “I found it difficult to relax” and “I felt that I was using a lot of nervous energy”). The DASS-21 has the ability to distinguish depression and anxiety, which are often highly correlated in other scales, so the depression and anxiety subscales were selected for use in this study [99]. Participants were asked to rate how often each statement applied to themselves on a scale of “0” (did not apply to me at all) to “3” (applied to me very much, or most of the time). Higher scores reflect more severe depression and anxiety. The DASS-21 is a reliable and valid measure assessing features of depression, anxiety, and tension/stress and has several advantages over the 42-item version, such as having a cleaner factor structure that makes it more easily interpreted than the original DASS [100].

*Psychological health subdomain: Perceived stress.* The 10-item self-reported Perceived Stress Scale (PSS-10 [87]) was used to assess perceived stress. Participants were asked to indicate how often, within the last month, they felt/thought about a series of statements on a scale of “0” (never) to “4” (very often). The total scores range from 0 to 40. Higher scores indicate greater severity of perceived stress. The PSS-10 has acceptable psychometric properties that are superior to those of the PSS-14 and has been recommended for measuring perceived stress in both research and clinical practice [101].

*Psychological health subdomain: Psychological Health.* The 42-item self-reported Lifestyle and Habits Questionnaire-Brief (LHQ-B) was also used to assess individuals’ psychological health via the psychological health subscale. Scores ranges from 0 to 35. Higher scores reflect greater perceived psychological health.

*Psychological health subdomain: Negative and positive affect.* The 20-item self-reported Positive and Negative Affect Schedule (PANAS-SF [102]) was used to assess how often individuals have felt a number of emotions. Participants were asked to indicate the extent to which they have felt each emotion on a scale of “1” (very slightly or not at all) to “5” (extremely). Subscale scores range from 20 to 100. Higher scores reflect higher levels of positive and negative affectation. Example items include “Interested,” Irritable,” “Jittery,” and “Ashamed.” The PANAS-SF has demonstrated high internal consistency and convergent and discriminant validity [102].

*Psychological health subdomain: Attenuated negative psychotic symptoms.* The 15-item Motivation and Pleasure Scale-Self-Report (MAPS-SR-15 [103]) was used to assess the motivation and pleasure domains of negative symptoms. Participants were asked how much pleasure they have experienced from a number of activities in the past week on a scale of “0” (no pleasure) to “4” (extreme pleasure), followed by how often they have experienced pleasure from that activity on a scale of “0” (not at all) to “4” (very often). The total scale score ranges from 0 to 60. Higher scores reflect greater severity of negative symptoms. The MAPS-SR has demonstrated good internal consistency and convergent and discriminant validity [103].

*Psychological health subdomain: Attenuated positive psychotic symptoms.* A modified 21-item self-reported Prodromal Questionnaire-Brief Version (PQB-21 [104]) was used to assess the frequency of attenuated positive psychotic symptoms experienced by participants over the last thirty days on a Likert scale from “0” (never) to “4” (daily), while not under the influence of drugs, alcohol, or other medication. This measure has been modified from the original version that examines the same psychotic-like experiences and related distress using a true/false question format to present the statement versions of items that were used in the original 92-item Prodromal Questionnaire-Likert Scale version to capture frequencies of the experiences endorsed by participants. If a participant reported having an experience in the past thirty days, they were presented with a follow-up question, “When this happens, how frightened, concerned, or distressed do you feel?” which was also answered on a Likert scale from “0” (not at all) to “4” (extremely). For the purpose of this study, we used a continuous quantitative frequency score to capture the rate at which participants reported experiencing attenuated positive psychotic symptoms over the past thirty days. The total scores range from 0 to 84. Higher scores reflect higher experience of attenuated positive psychotic symptoms. The original Prodromal Questionnaire (PQ-92) is the most commonly used psychosis risk-screening instrument in the literature [105]. The incorporation of distress in this measure improves the accuracy of screening by minimizing false positives [106]. Results from validity studies on the PQB-21 found it to be comparable to the PQ-92, deeming it a good self-reported screener for selecting subjects to interview for psychosis risk in secondary mental health care services [107]. The PQ-B single factor showed adequate goodness of fit index, and the total frequency score demonstrated excellent internal consistency in community-derived adolescents [99,108].

*Psychological health subdomain: Attention and concentration.* The 6-item Adult ADHD Self-Report Screening Scale for DSM-5 (ASRS-5 [109]) was used to assess attention and concentration. Participants were asked to rate how often they have felt and conducted themselves in a series of ways over the past 30 days on a scale of “1” (never) to “5” (very often). A continuous quantitative total summed scale score was created for the purpose of examining participants’ levels of impairment in attention and concentration. The total scale scores range from 0 to 74 based on the instrument’s assigned standardized scoring instructions. Higher scores reflect greater impairment in attention and concentration. The ASRS-5 has demonstrated excellent operating characteristics in classifying persons that met diagnostic criteria for ADHD in general population and special treatment samples [109].

*Psychological health subdomain: Working memory and inhibition.* The 14-item self-reported Adult Executive Functioning Inventory (ADEXI-14 [110]) was used to assess executive functioning, with a particular focus on working memory and inhibitory control. Participants are asked to rate how much each statement describes them on a scale of “1” (definitely not true) to “5” (definitely true). Subscale scores range from 5 to 45. Higher scores reflect greater working memory and inhibition. The ADEXI has demonstrated high internal consistency and adequate test–retest reliability [110].

*Psychological well-being subdomain*. Psychological well-being has long been understood to be a multidimensional component of quality of life [111,112,113]. The dimensions include perceptions of one’s own health, socialization, a sense of autonomy and mastery over one’s immediate environment, a sense of ongoing personal growth, and self-acceptance.

*Psychological well-being subdomain: Psychological well-being.* The widely used 18-item self-reported Psychological Well-being Scale (PWS-18 [112]) was used to assess six dimensions of psychological well-being, including autonomy, environmental mastery, personal growth, positive relationships with others, purpose in life, and self-acceptance. Participants were asked to rate how strongly they agree or disagree with a series of statements on a scale of “1” (strongly agree) to “7” (strongly disagree). Subscale scores range from 3 to 21. Higher scores reflect higher levels of psychological well-being. The Ryff Scales of Psychological Well-Being have been found to be a valid and reliable measure of psychological well-being [114,115]. However, the “Purpose in life” subscale of this instrument failed to meet reliability criteria for this study and was excluded from further analysis.

*Psychological well-being subdomain: Participation limitations.* The 13-item self-reported *Participation Scale-Short Form* (PS-SF [113]) was used to assess limitations and restrictions on social participation. This measure is commonly used in rehabilitation, stigma reduction, and social integration programs. Participants were asked to respond to a series of questions about their social participation, including how big of a problem that area is to the person on a scale from “1” (no problem) to “5” (large). Total scores range from 0 to 90. Higher scores reflect greater restriction. The PS-SF demonstrated good validity and internal consistency and was highly correlated with the full version of the Participation Scale, which supports its construct validity [113].

*Psychological well-being subdomain: Health-related quality of life.* The 14-item self-reported CDC “Health Days Measure” (CDC HRQOL [111]) was used to assess individuals’ perceived physical and mental health, including a subjective rating of their general health, number of bad physical and mental health days, and number of days that their health contributed to their inability to partake in routine activities over the past thirty days. Participants were asked to answer questions related to their health and its effects on their routines. The HRQOL has demonstrated construct, criterion, internal, and concurrent validity and test–retest reliability [110].

*Social connectedness subdomain*. Social connectedness refers to persons’ general sense of belonging and interpersonal closeness in their social world [116]. For the purposes of this study, social disconnectedness is presumed to reflect the collective influence of social-cognitive and emotional self-regulation deficits on a person’s global social functioning, especially the condition of his/her social support system.

*Social connectedness subdomain: Loneliness.* The 20-item self-reported Revised UCLA Loneliness Scale Version 3 (UCLA-R V3 [117]) was used to assess subjective feelings of loneliness and social isolation. Participants were asked to indicate how often each of the statements was descriptive of them on a scale of “1” (never) to “4” (often). The total scores range from 20 to 80. Higher scores reflect greater feelings of loneliness. Example items include, “How often do you feel that you are ‘in tune’ with the people around you?”; “How often do you feel that you lack companionship?”; and “How often do you feel left out?” This version of the measure has been found to have high reliability in both internal consistency and test–retest reliability over the period of one year, along with convergent validity with other measures of loneliness [117].

*Social connectedness subdomain: Social connectedness*. The 20-item self-reported Social Connectedness Scale-Revised (SCS-R [116]) was used to assess individuals’ general sense of belonging and interpersonal closeness with the social world. This measure was selected because many other widely used measures of social functioning include behavioral items that were not appropriate given the context of the ongoing COVID-19 pandemic. Participants rated their agreement with items on a 6-point Likert-type scale. The total scores range from 20 to 120. A higher total score reflects a greater degree of social connectedness. Example items include “Even among my friends, there is no sense of brother/sisterhood” and “I don’t feel related to anyone.” The SCS-R has demonstrated good convergent and divergent validity, and internal consistency [116].

*Social connectedness subdomain: Social support*. The 12-item Multidimensional Scale of Perceived Social Support (MSPSS [118]) was used to assess individuals’ perceived adequacy of social support from three sources: friends, family, and a significant other. Participants were asked to rate how strongly they agreed or disagreed with each statement on a scale of “1” (very strongly disagree) to “7” (very strongly agree). Subscale scores range from 12 to 84. Higher scores reflect greater perceived adequacy of social support. Example statements include “There is a special person around when I am in need,” “My family tries to help me,” “I get the emotional help and support I need from my family,” and “I can count on my friends when things go wrong.” The SCS-R has demonstrated adequate internal and test re-test reliability, strong factorial validity, and moderate construct validity [118].

### 2.3. Data Analysis

All data analysis was conducted with SPSS version 28.0.0.0.

A principal axis factor analysis was performed for all the selected measures within each domain or subdomain. For the measures that generate subscales, all subscales that met reliability criteria were included, but scales representing a total or summary score for a single measure with multiple subscales were excluded (this was to reduce the influence on the factor solutions of correlated subscales of a single instrument). There were thus 8 factor analyses, reflecting the developmental domain; the psychological resilience and healthy habits subdomains; the physical health, psychological health, and sleep subdomains; the psychological well-being subdomain; and the social well-being subdomain. The standard criteria for factor extraction were applied, an eigenvalue >1.0, with a maximum of 25 iterations for convergence. The extracted factors were subjected to direct oblimin rotation, to allow for expected correlations between factors.

The resulting factors were arranged in an a priori sequence of causal influence across domains, from developmental vulnerabilities to resilience, to health, to quality of life. The participants’ computed factor scores were then subjected to simple linear multiple regression analyses, wherein each factor score in turn was the dependent or “target” variable (except for the developmental factors, assumed to be exogenous variables), and the factors in the sequentially prior domain were the independent variables. To control for randomly missing data and non-normal distributions [119], parameter estimations for the linear regressions were produced by bootstrapping (nonparametric resampling) 10,000 resamples with replacement.

## 3. Results

### 3.1. Demographics

The demographics and other sample characteristics are shown in Table 1. The demographics are consistent with expectations for a large public university in the U.S. Great Plains. As expected, due to the stratified recruitment strategy, students with histories of chronic medical conditions, mental health treatment, and disabilities are disproportionately represented.

### 3.2. Factor Analyses

The results of the eight factor analyses across domains and subdomains are summarized in Table 2. A total of 17 factors were extracted. For clarity of interpretation, the valences of correlations and loadings are shown as positive to indicate significant and/or meaningful correlations in the expected direction, i.e., fewer vulnerabilities being associated with better health (in the initial output, the valences of the various correlations and loadings were mixtures of positive and negative, because higher numbers reflect lower vulnerability or better health for some measures, and the reverse for others). There were no significant and/or meaningful valences in the unexpected or counterintuitive direction, and therefore, no negative valences are shown in the table. Generally, as expected, the first factor identified in the analyses accounted for 25% to 75% of the total variance within the domain or subdomain and included all or most of the measures.

*Developmental domain.* The 16 variables entered yielded an unrotated factor solution that accounts for 53.2% of the variance within the domain. Three factors were identified. The unrotated factors account for 30.4%, 12.7%, and 10.0% of the variance. Based on the rotated structure matrix, the respective factors were labeled *overall adversity and vulnerabilities*, *trauma and adversity*, and *vulnerability sans adversity*. The rotated factors are modestly intercorrelated, from r = 0.23 to r = 0.30.

*Psychological resilience subdomain*. The 20 variables entered yielded an unrotated factor solution that accounts for 52.9% of the variance within the subdomain. The unrotated factors account for 33.9%, 11.7%, and 7.3% of the variance. Three factors were identified. Based on the rotated structure matrix, the respective factors were labeled *overall resilience*, *worried metacognition*, and *unreflective action*. The rotated factors are modestly intercorrelated, from r = 0.18 to r = 0.28.

*Healthy habits subdomain.* The 12 variables entered yielded an unrotated factor solution that accounts for 49.2% of the variance within the subdomain. The unrotated factors account for 25.0%, 12.5%, and 11.8% of the variance. Three factors were identified. Based on the rotated structure matrix, the respective factors were labeled *overall healthy habits*, *smoking and drinking*, and *sweet tooth*. There are no meaningful correlations between the factors.

*Physical health subdomain*. The three variables entered yielded an unrotated factor solution that accounts for 76.6% of the variance within the subdomain. Only one factor was identified. Based on the unrotated factor matrix, it was labeled *overall physical health*.

*Sleep subdomain*. The three variables entered yielded an unrotated factor solution that accounts for 64.0% of the variance within the subdomain. Only one factor was identified. Based on the unrotated factor matrix, it was labeled *sleep quality.*

*Psychological health subdomain.* The 11 variables entered yielded an unrotated factor solution that accounts for 74.6% of the variance within the subdomain. Three factors were identified. The unrotated factors account for 53.3%, 12.0%, and 9.4% of the variance. Based on the rotated structure matrix, the respective factors were labeled *overall psychological health*, *cognitive difficulty*, and *affective activation*. The only meaningful correlation between factors was that of overall psychological health with cognitive difficulty (r = 0.48).

*Psychological well-being subdomain*. The 11 variables entered yielded an unrotated factor solution that accounts for 59.0% of the variance within the subdomain. Two factors were identified. Based on the rotated structure matrix, the factors were labeled *overall well-being* and *health-related restrictions*. The unrotated factors account for 14.6% and 7.8% of the variance, respectively. The two factors are meaningfully correlated (r = 0.44).

### 3.3. Regression Analyses

The results of the sequenced linear regression analyses are summarized in Figure 1. The three factors in the developmental subdomain significantly predicted (in the statistical sense; *p* < 0.05 for the R^2^ of the equation) four of the six factors in the behavioral and psychological resilience domain. Four of the six behavioral and psychological resilience factors contributed to prediction of all five of the factors in the health domain. Four of the five health factors contributed to prediction of all three quality of life domains. The strength of the collective predictions for the respective target factors highly varied, ranging from 0% to 79% of the variance of the target factors. For the significantly predicted target factors, one to four of the factors in the adjacent domain contributed uniquely to the prediction.

To summarize the results, the factor analyses showed that common sources of variance, factors, could be identified within the domains and subdomains that do not simply reflect specific measures or the constructs or paradigms on which they are based. The sequential linear regression analyses showed that there are significant relationships between factors in adjacent domains.

## 4. Discussion

The factor analyses were successful in identifying common sources of variance across different measures, derived from different constructs and paradigms, within their respective domains and subdomains. This supports the feasibility of a research approach that posits a generalized progression across multiple domains of biosystemic functioning, from vulnerability states at birth or acquired in early childhood, to individual differences in resilience, to health disparities, to compromised quality of life.

The regression analyses identified the overall psychological resilience factor as associated with all five factors in the health domain; there were additional links to two or more health domain factors by four of the six factors in the resilience domain. Overall resilience was also fairly strongly associated with all three of the developmental factors, suggesting it may represent an important common pathway between vulnerability and health disparities.

Healthy habits are weakly associated with affective activation and sleep quality, and not at all with physical health. Smoking and drinking are not linked to any of the health factors. This would seem to be counterintuitive but may reflect a key feature of the university population, youthfulness. They may be too young for bad habits to affect their health, yet. This may be an important consideration for continued research on health disparities in vulnerable university populations. Additionally, although sleep quality is associated with healthy habits, overall resilience, and worried metacognition, it is not in turn linked to any quality-of-life factors. This may reflect a leveling effect of the myriad sources of sleep irregularity and disruption in student life.

Physical health and psychological health are associated with different factors in the quality of life domain. Physical health in this population is associated with health-related restrictions on desirable and important activities, and with social connectedness, but not with overall well-being, which is more strongly associated with psychological health factors. This may also be a feature of vulnerability in university students, who presumably have better access to basic healthcare than other vulnerable populations. Additionally, the culture and priorities of higher education provide sources of satisfaction and opportunities for achievement relatively insulated from health problems—e.g., being a successful student—which are less available in other environments. In less supportive environments, especially where poverty is more ubiquitous, the impact of physical ill health should be expected to be less buffered. Nevertheless, it is important that even in the supportive university environment, poor health is still associated with activity restrictions and social disconnectedness.

The pathways revealed by the regression analyses indicate that processes in the intermediate domains identified in this study participate in the impacts of vulnerability on health, quality of life, and social connectedness, but this is not an exhaustive account of all possible pathways between developmental vulnerabilities, resilience factors, health, and quality of life, even among the factors identified in this study. A larger-scale longitudinal dataset using these variables will allow simultaneous evaluation of all possible pathways between the factors identified in this study, including “direct” effects of each factor upon the others, and “indirect” effects, mediated or moderated by intermediate factors. For example, such an analysis would be expected to identify a “direct” effect of developmental vulnerabilities upon quality of life, in addition to the “indirect” effects identified in this analysis, via resilience and health. For practical purposes, the mechanisms of such “direct” effects are unknown. They presumably consist of multiple “indirect” pathways, of which the pathways identified in this study are a subset. In that sense, progress in understanding the role of vulnerability in health disparities will be through the identification of all the “indirect” pathways between the two. This could include domains and subdomains not considered in the present study.

### 4.1. Limitations and Strengths

This study’s limitations include its cross-sectional nature, the sample’s use of college students with an over-representation of disabilities corresponding to recruitment procures, and use of self-reported data. While the cross-sectional nature of the study (data examined at baseline timepoint) renders the path model results as illustrating preliminary findings on various domains of vulnerability, resilience, health habits, and mental and physical health for informing health disparities research, it is within the context of a large-scale longitudinal study with ongoing data collection. The longitudinal study will allow for the follow-up examination of domains identified in this study via longitudinal structural equation modeling, which could provide us with more information on direct and indirect effects and allow us to disaggregate into between-person (i.e., person mean across the study; Level 2) and within-person components (i.e., deviation from the person mean at a given time point; Level 1) [120]. Additionally, the present study employed an analytic design that measured the included constructs in a temporal, sequential, and developmental sequence.

Though some may find the use of a college student sample limiting, it is but a minor limitation, as no matter what population, the same procedures would be used. Moreover, this study’s unique recruitment procedures allowed for sample stratification by physical health status, such that approximately half of the participants reported having at least one physician-diagnosed chronic health condition. This enriched sample comprised of both students with chronic health conditions and an otherwise physically healthy comparison group provided helpful data for an examination of health disparities in a non-clinical, emerging adult sample. While there would be no reason that the findings would generalize to other populations, college students are not merely a “convenience sample” for schizotypy research, especially in the context of health behaviors and outcomes. Adolescence and early adulthood represent the peak age of onset for psychosis: 80% of first-episode psychoses occur between the ages of 16 and 30 [121,122]. Sleep disturbance is more prevalent during this period, as demonstrated by 36% of college students reporting significant sleep disturbances [123,124,125]. Thus, college students are of interest due to their demographics, developmental stage, range of personal/social functioning, health behaviors, and other characteristics, which highlight the value of better understanding how these dimensions play out in this population sample and how they work differently in others. As within all areas of science, best practice is to incorporate both non-clinical and clinical population samples to better understand theoretical progress, and models of vulnerability and resiliency to inform research on health disparities.

This study’s unique methodological approach established a set of canonical variables commonly utilized in vulnerability and health disparities research that may be suitable for examining processes across multiple domains of biosystemic functioning and progressive stages of human development. Thus, the research helped to identify some of the common elements across multiple paradigms and constructs that create pathways spanning from genetic and acquired vulnerabilities, to behavioral and lifestyle patterns contributing to risk and resiliency in health, and ultimately to a person’s quality of life.

### 4.2. Future Directions

Consideration of existing findings on developmental vulnerabilities led to psychological constructs whose developmental courses are at least partially understood. Such constructs are leading candidates for revealing common factors that ultimately impact health disparities and quality of life. Eventually, large-scale longitudinal studies of vulnerable populations will be necessary to fully understand all the mechanisms of the effects of psychiatric disorders on health disparities, and in turn, the effects of health on quality of life. Meanwhile, identification of the underlying dimensions of those mechanisms, as revealed by canonical factors of vulnerability, resilience, health, and quality of life, provide important clues about the pathways of particular vulnerabilities toward the real-world consequences we observe.

Thus, research must evolve in the direction of increasingly complex multivariate analyses capable of evaluating multiple domains at different stages of development. One strategy to progress from focused studies of particular constructs and processes to the bigger picture in the real world is to empirically consolidate previously developed constructs into manageable sets of canonical measures that reflect their key common components, within broader domains of biosystemic functioning, with relevance to health disparities and the factors that influence them.

## 5. Conclusions

The approach to vulnerability and health disparities taken in this study complements studies of more particular paradigms and constructs. For example, in a recent larger-scale cross-sectional study in our research group [39], a focused analysis of the three dimensions of schizotypy (positive, negative, and disorganized) revealed different pathways to social connectedness, via developmental and psychological variables. The approach of the present study provides a broad perspective on the involvement of major domains of measurement and biosystemic functioning, whereas a focus on specific constructs within those domains reveals particular causal mechanisms. Ultimately, an understanding of causal relationships between particular constructs will be necessary for designing interventions to reduce or prevent the impact of vulnerability and other aspects of mental health on health disparities. Meanwhile, the approach of this study will helpfully identify all the domains and particular constructs necessary to achieve an exhaustive and comprehensive account, and accordingly, a complete intervention armamentarium.

## Figures and Tables

**Figure 1 behavsci-12-00240-f001:**
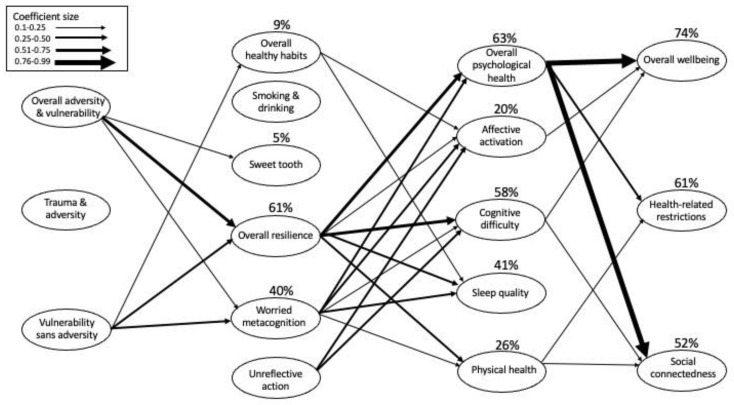
Pathways across domains identified by multiple regressions; percent figures indicate variance accounted for (R^2^ of the multiple regression solution) by factors in the previous domain.

**Table 1 behavsci-12-00240-t001:** Sample demographics (N = 213).

**Age (years), mean (SD) [range]**	20.530 (2.804) [15]
**Sex assigned at birth, N (%)**	
Female	173 (81.2%)
Male	40 (18.8%)
**Race, N (%)**	
White or European America	177 (83.1%)
Black or African American	4 (1.9%)
Asian American	18 (8.5%)
American Indian or Alaska Native	2 (0.8%)
Native Hawaiian/Pacific Islander	2 (0.9%)
Other	10 (4.7%)
**Ethnicity Hispanic/Latinx, N (%)**	16 (7.5%)
**Relationship status, N (%)**	
Single, never married	144 (67.6%)
Dating	53 (24.9%)
Committed relationship	67 (31.4%)
Married without children	2 (0.9%)
Married with children	2 (0.9%)
**Average childhood household SES, N (%)**	
Less than $25,000	12 (5.7%)
$25,000 to $34,999	8 (3.8%)
$35,000 to $49,999	22 (10.4%)
$50,000 to $74,999	38 (17.9%)
$75,000 to $99,999	38 (17.9%)
$100,000 to $149,999	48 (22.6%)
$150,000 or more	46 (21.7%)
**Urbanicity, N (%)**	
Rural	48 (22.5%)
Suburban	122 (57.3%)
Urban	43 (20.2%)
**Medical History, N (%)**	
Physician Diagnosed Chronic Health Condition	107 (50.2%)
Physician Diagnosed Sensory Impairment	87 (40.8%)
**Psychiatric History, N (%)**	
Lifetime Psychotherapy	104 (48.8%)
Lifetime Inpatient Treatment	10 (4.7%)
Current Psychotropic Medication Use	72 (33.8%)
First-degree Genetic Family Autism Spectrum Disorder	10 (4.7%)
Second-degree Genetic Family Autism Spectrum Disorder	24 (11.3%)
First-degree Genetic Family Schizophrenia Spectrum/Psychotic Disorder	4 (1.9%)
Second-degree Genetic Family Schizophrenia Spectrum/Psychotic Disorder	7 (3.3%)
**Participant Identified Disability, N (%)**	
No Disability (None)	114 (53.5%)
Medical/physical Disability	40 (18.8%)
Neurological Disability	21 (9.9%)
Cognitive	19 (8.9%)
Mental Health	65 (30.5%)
Other	2 (0.9%)

**Table 2 behavsci-12-00240-t002:** Computed factors and loadings.

** Developmental domain **	** loadings **
**Factor 1. Overall adversity and vulnerabilities: 30.4% ***	
Alexithymia Subscale “Describing”	0.76
Alexithymia Subscale “Identifying”	0.69
Schizotypy Subscale “Disorganized”	0.65
Alexithymia Subscale “Externally oriented”	0.49
Childhood trauma Subscale “Emotional abuse”	0.37
Childhood trauma Subscale “Emotional neglect”	0.37
Schizotypy Subscale “Positive”	0.37
Distress tolerance Subscale “Appraisal”	0.30
Distress tolerance Subscale “Absorption”	0.26
**Factor 2. Trauma and adversity: 12.7% ***	
Childhood trauma Subscale “Emotional abuse”	0.86
Childhood trauma Subscale “Emotional neglect”	0.79
Childhood trauma Subscale “Physical neglect”	0.73
Childhood trauma Subscale “Physical abuse”	0.68
Childhood household income	0.39
Schizotypy Subscale “Disorganized”	0.39
Distress tolerance Subscale “Absorption”	0.30
Alexithymia Subscale “Describing”	0.30
Alexithymia Subscale “Identifying”	0.30
Schizotypy Subscale “Positive”	0.25
Distress tolerance Subscale “Appraisal”	0.25
**Factor 3. Vulnerability sans adversity: 10.0% ***	
Distress tolerance Subscale “Absorption”	0.88
Distress tolerance Subscale “Appraisal”	0.87
Alexithymia Subscale “Identifying”	0.57
Distress tolerance Subscale “Regulation”	0.56
Schizotypy Subscale “Disorganized”	0.49
Alexithymia Subscale “Describing”	0.37
Schizotypy Subscale “Positive”	0.37
Childhood trauma Subscale “Emotional abuse”	0.33
Childhood trauma Subscale “Emotional neglect”	0.32
** Behavioral and psychological resilience domain **	
*Psychological resilience subdomain*	
**Factor 4. Overall resilience: 33.9% ***	
Sense of purpose Subscale “Awareness”	0.77
Self-esteem Scale	0.75
GRIT	0.74
Self-efficacy	0.69
Self-concept clarity	0.68
Sense of purpose subscale “Awakening”	0.66
Mindfulness Subscale “Acting with awareness”	0.63
Mindfulness Subscale “Describing”	0.62
Metacognition Subscale “Lack of confidence”	0.57
Internal powerful other Subscale “Internality”	0.53
Internal powerful other Subscale “Chance”	0.52
Internal powerful other Subscale “Powerful other”	0.52
Mindfulness Subscale “Nonjudgmental”	0.50
Sense of purpose Subscale “Altruistic purpose”	0.45
Metacognition Subscale “Negative beliefs”	0.44
Metacognition Subscale “Nonreactivity”	0.38
Metacognition Subscale “Need to control”	0.26
**Factor 5. Worried metacognition: 11.7% ***	
Metacognition Subscale “Negative beliefs”	0.66
Metacognition Subscale “Need to control”	0.65
Mindfulness Subscale “Nonjudgmental”	0.62
Self-Concept Clarity	0.56
Self-esteem Scale	0.49
Metacognition Subscale “Self-consciousness”	0.49
Metacognition Subscale “Positive beliefs about worry”	0.39
Internal powerful other Subscale “Powerful other”	0.36
Metacognition Subscale “Lack of confidence”	0.35
Mindfulness Subscale “Describing”	0.33
Internal powerful other Subscale “Chance”	0.32
Metacognition Subscale “Nonreactivity”	0.28
**Factor 6. Unreflective action: 7.3% ***	
Mindfulness Subscale “Observing”	0.58
Metacognition Subscale “Self-consciousness”	0.48
Metacognition Subscale “Nonreactivity”	0.46
Self-efficacy	0.32
Sense of purpose Subscale “Altruistic purpose”	0.28
Self-esteem Scale	0.27
Mindfulness Subscale “Acting with awareness”	0.26
*Behavioral resilience subdomain*	
**Factor 7. Overall healthy habits: 25% ***	
LHQB Subscale: nutrition	0.73
Diet: veggies	0.70
Diet: fruit	0.65
diet: leafy greens	0.61
diet: grains	0.52
LHQB Subscale: Health & exercise	0.50
**Factor 8. Smoking & drinking: 12.5% ***	
nicotine use	0.68
alcohol consumption	0.40
cannabis use	0.35
**Factor 9. Sweet tooth: 11.8% ***	
diet: sugary foods	0.50
LHQB Subscale: nutrition	0.42
diet: fruit	0.32
diet: dairy	0.31
** Health and illness domain **	
*Physical health subdomain*	
**Factor 10. Physical health: 63.1% ***	
CHIPS physical symptom distress	0.90
Physical symptoms report	0.87
Sickness questionnaire	0.65
*Sleep subdomain*	
**Factor 11. Sleep quality 64% ***	
Daytime Insomnia symptoms scale	0.69
Insomnia severity index	0.68
Ford stress insomnia scale	0.67
*Psychological health subdomain*	
**Factor 12. overall psychological health: 53.3% ***	
DASS Subscale: “Depression”	0.89
PSS Subscale: perceived stress	0.85
LHQB Subscale: “Psychological health”	0.80
DASS Subscale: “Anxiety”	0.78
PANA subscale: “Negative affect”	0.76
PANA subscale: “Positive affect”	0.75
MAPS total psychopathology	0.74
PQB attenuated psychotic symptoms	0.60
ADHD screener	0.54
AEFI Subscale: “Working memory”	0.45
AEFI Subscale: “Inhibition”	0.30
**Factor 13. Cognitive difficulty: 12.0% ***	
AEFI Subscale: “Working memory”	0.88
AEFI Subscale: “Inhibition”	0.87
ADHD screener	0.54
PQB attenuated psychotic symptoms	0.54
LHQB Subscale: “Psychological health”	0.51
PSS Subscale: perceived stress	0.46
DASS Subscale: “Depression”	0.45
PANA subscale: “Negative affect”	0.42
PANA subscale: “Positive affect”	0.42
MAPS total psychopathology	0.28
**Factor 14. Affective activation: 9.4% ***	
PQB attenuated psychotic symptoms	0.51
PANA subscale: “Positive affect”	0.49
PANA subscale: “Negative affect”	0.45
DASS Subscale: “Anxiety”	0.45
MAPS total psychopathology	0.43
** Quality of life domain **	
*Psychological well-being subdomain*	
**Factor 15. overall wellbeing: 44.0% ***	
Wellbeing Subscale: “Self-acceptance”	0.84
Wellbeing Subscale: “Mastery 1”	0.73
Participation scale	0.70
Wellbeing Subscale: “Personal growth”	0.68
Wellbeing Subscale: “Purpose in life”	0.66
Wellbeing Subscale: “Personal relations”	0.65
Bad mental health days	0.49
Wellbeing Subscale: “Mastery 2”	0.47
General health self-rating	0.43
Days of health restrictions	0.42
**Factor 16. Health-related restrictions 14.6% ***	
Days of health restrictions	0.82
Bad physical health days	0.72
Bad mental health days	0.68
General health self-rating	0.64
Wellbeing Subscale: “Mastery 1”	0.54
Wellbeing Subscale: “Self-acceptance”	0.52
Participation scale	0.46
Wellbeing Subscale: “Personal relations”	0.28
*Social wellbeing subdomain*	
**Factor 17. Social connectedness: 63.1% ***	
UCLA loneliness scale	0.95
Social connectedness scale	0.90
MSPSS Subscale: “Friends”	0.73
MSPSS Subscale: “Family”	0.56
MSPSS Subscale: “Significant other”	0.51

* Percent of variance accounted for by factor within the domain or subdomain.

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
