# Peer review of "Domains of Vulnerability, Resilience, Health Habits, and Mental and Physical Health for Health Disparities Research"

_behavsci, 2022, doi:10.3390/bs12070240_

Round 1

Reviewer 1 Report

The manuscript “Domains of vulnerability, resilience, health habits, and mental and physical health for health disparities research” presents an important topic of examining health disparities among people with mental issues. In my opinion this manuscript adds important information to the field of mental health and quality of life literature. 

However, there are some major as well as minor issues throughout the text, which I will list in order.

Firstly, to be honest upon reviewing this manuscript I found it difficult to concentrate on myriad measures, domains, subdomains, etc... While it is important to include all the variables of interest in the manuscript the abundance of measures can be overwhelming to the reader. But that is my opinion, others might disagree with me. 

Secondly, Introduction should be more concise focusing on the previous research, background and stating the purpose of the current study, instead of including the definitions of some of the measures in this section. More detailed comments on Intro: 

1) Maybe elaborate a little after using the term SMI, like listing what illness is considered to be "serious" 

2) In my opinion, the lines 66 through 72 are more suitable for Discussion section where the authors state their future goals.

3) Same comment about line 89-106, probably would be better in the Discussion 

4) In my opinion , Lines 106 through 262 are more suitable for Methods section.

5) In Data Analysis section:  line 673-675 mention of "dependent variable" is repeated. Change to "independent" ?

Regarding Tables: while one appreciates the display of all the results it can be a bit overwhelming since the Tables appear to be long and "busy"
I wonder if there is a way to make them a little more "readable" by making the font less and stretching the rows so that the text in one cell fills only one row instead of taking too much space making the table look "messy"

In Table 1 only Ethnicity is in Italics - change 

In Table 2 be consistent with using Bold font. Would advise to use Bold only for the titles or definitions and getting rid of Bold for listing subdomains and subscales.

Finally, the manuscript lacks a whole section on Strengths and Limitations of the study in Discussion section. Would make the manuscript more effective including those. 

I found this manuscript interesting and worth publishing but it needs a little bit of polishing before doing so.

Author Response

Dear Reviewer,

            We thank you for your thoughtful contributions to the editing of our manuscript. Please see below for our response to your comments:

Organization/manuscript business: We addressed this problem by separating an Introduction Section and a Background Section. We also moved some of the subdomain information into the methods section to organize it in close succession with the group of measures it is describing.

Item 1: We added a definition for serious mental illness into the introduction where we introduced the term to clarify. We re-organized the paper and restructured the introduction safer to make it more concise. See comment above.

Item 2: We moved lines 66 through 72 to the “future directions” section of the discussion.

Item 3: We moved lines 89 – 99 to the discussion section. We kept part of 99 – 106 with the domain descriptions, which were moved to the measurement portion of the methods section. Organizationally, we think that part of the content defining and providing information on schizotypy would be helpful to keep with the domain/construct information and present earlier on in the manuscript than in the discussion section.

Item 4: We moved this section describing the different subdomains to the measurement portion of the methods section.

Item 5: We corrected this typo to “independent” and added alternate “predictor variables” to be symmetric with “dependent” and “target”

Item 6: We must defer to the editors regarding table formats and will work with them to optimize readability – we agree that the way it comes out in the template is not optimal.

Item 7 (Table 1): Please see above comment re table format. We changed this line from italics to bold font to be consistent with the rest of the headings. I also changed the age line for consistency. We updated tables into two columns to make them less busy. Rows are stretched and text only fills one cell.

Item 8 (Table 2): Please see above comment re table format.

Item 9: We added a strengths and limitations section to the discussion.

Item 10: We did another round of revisions throughout the manuscript.

                                                                                                                                                                  Kind regards,

                                                                                                                                                                  Rebecca Wolfe

Reviewer 2 Report

This is a good paper focusing on health disparities in people with severe mental illness within the point of view of public health concerns. With the starting idea that patients with severe mental illness suffer from premature death, the authors aimed to identify causes of disparities and differences in health-related behaviors and resilience in public university students.

The paper is well-written and is of interest for the readers and the journal. However, several minor changes are recommended before considering before its publication.

In the introduction section, the authors report in the second paragraph that health disparities create the risk or have an effect on the onset of a certain disorder. This should be better explained. Health disparities probably influence the onset or worsening of mental health symptoms, and viceversa; the worsening or influence of mental health on health is also important. I would recommend to talk about both separately. 

In the second and third paragraph there is a lack of references supporting the ideas. In fact, in the third paragraph, the authors report that most previous research has been within specific theoretical context, for instance, the construct of schizotypy. I recommend to add several references.

At the end of the introduction section the authors report that "four domains ... were initially identified". In the introduction section, the authors cannot start describing results from the study.

The introduction is too long. I would recommend to include some aspects into the discussion section.

Plese, number subsections of the methods section as: 2.1. Participants and procedures, 2.2. Measures, etc.

The discussion section is brief. I would recommend to expand it with some lines of the introduction to better discuss the main findings.

I recommend to add a conclusions sections and a subsection about limitations and strenghts.

Author Response

Dear Reviewer,

            We thank you for your thoughtful contributions to the editing of our manuscript. Please see below for our response to your comments:

Item 1: We do not intend to suggest that health disparities create the risk or have an effect on the disorder beyond the effects of the health problems themselves.  We have revised the text to clarify this.

Item 2: Appropriate references were added to the schizotypy paragraph in the intro.

Item 3: We failed to sufficiently distinguish between our description of previous studies and our current hypotheses - our revisions clarify the distinction. The term “identified” here refers to what emerges from our Intro, not a formulation of new hypotheses, and is a framework for guiding instrument selection and organizing the analyses. Clarifying language was also added to this background section.

Item 4: Content was moved from the intro to the methods and discussions sections of the manuscript. It was also re-organized with subheaders.

Item 5: Numbers were added for the subsections including 2.1. Participants and procedures, 2.2. Measures, 2.3 Data analysis, 3.1 Demographics, 3.2 Factor analyses, 3.3 Regression analyses, 4.1 Study strengths, and limitations, 4.2 Future directions

Item 6: The discussion section was expanded. Also, some content from the intro was moved into the discussion section.

Item 7: Subsections for strengths, limitations, and conclusions were added to the discussion section.

                                                                                                                                                                  Kind regards,

                                                                                                                                                                  Rebecca Wolfe